# Towards a Standardized Representation for Deep Learning Collective Algorithms

Jinsun Yoo\*, William Won\*, Meghan Cowan†, Nan Jiang†, Benjamin Klenk†, Srinivas Sridharan†, Tushar Krishna\*,
\*Georgia Institute of Technology    †NVIDIA
\*{jinsun, william.won}@gatech.edu, tushar@ece.gatech.edu    †{mcowan, tedj, bklenk, srisridharan}@nvidia.com

*Abstract*—The explosion of machine learning model size has led to its execution on distributed clusters at a very large scale. Many works have tried to optimize the process of producing collective algorithm and running collective communication, which acts as a bottleneck to distributed machine learning. However, the lack of a standardized collective algorithm representation has hindered interoperability between the workload representation, collective algorithm producers, and consumers. The trend of collective algorithm producers and consumers using their own representation has pushed away from co-optimizing collective communications and the rest of the workload. Additionally, tool-specific conversions and modifications have to be made for each pair of tools producing and consuming collective algorithms.

In this paper, we propose a standardized workflow leveraging a common collective algorithm representation. Upstream producers and downstream consumers converge to a common representation format based on Chakra Execution Trace, which is being used to represent distributed machine learning workloads. Such a common representation enables to view collective communications at the same level as workload operations and decouple producer and consumer tools, enhance interoperability, and relieve the user from the burden of having to focus on downstream implementations. We provide a proof-of-concept of this standardized workflow by simulating collective algorithms generated by MSC-CLang domain-specific language through ASTRA-sim distributed machine learning simulator using various network configurations.

## I. INTRODUCTION

Recent trends in enormous machine learning (ML) models, such as recommendation models [1] or Large Language Models (LLMs) [2], [3], have made it impractical to execute them on a single Neural Processing Unit (NPU, such as GPU, TPU, or custom ASIC) [4]. Consequently, ML execution has evolved to *distribute* the job across multiple NPUs [5].

Within distributed ML, each of the participating NPUs completes a portion of the overall compute task. They periodically transfer and synchronize their intermediate compute results (e.g., weight or input gradients) in accordance with a predefined schedule [6]. Such traffic patterns are *collective* in nature. Therefore, *collective communication* primitives, such as All-Reduce or All-Gather, have been the key building blocks of distributed ML platforms [7].

The communication of intermediate data has become a bottleneck to the overall distributed ML execution, and recent studies have tried to optimize this [8]–[10]. Within the scope of collectives, Collective Communication Libraries (CCLs) such as NVIDIA's NCCL [11] provide implementations of several predefined *collective algorithms* (e.g., Ring [12] and Double

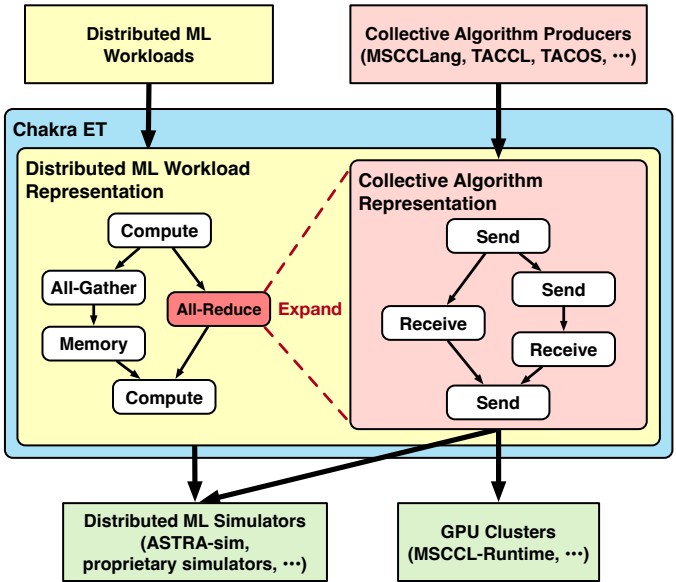

Fig. 1: The proposed standardized workflow using Chakra Execution Trace (Chakra ET) as a common representation for both distributed ML workload and collective algorithm. Downstream tools receive both workload and collective algorithms represented using a common Chakra ET format. Sample Chakra ETs of the workload (left) and that of the algorithm of a single collective (right) is also shown.

Binary Tree [13]). Several works have tried to further optimize collective algorithms [14], [15], by automatically *synthesizing* topology-aware algorithms [16], [17], or allowing users to easily define and test their own [18].

The outcome of such optimizations is not limited to actual execution on real clusters. For example, distributed ML simulators [19]–[21] allow users to swiftly compare the performance of different collective algorithms or workload parallelization strategies without running them on actual hardware. This enables them to easily test new optimizations for current systems and additionally allows the co-design and evaluation of futuristic distributed ML platforms.

**Unfortunately, these collective optimizations have been done separately from distributed ML workload optimization,** due to a lack of common representation. Specifically, distributed ML workload information does not contain details on how each collective operation is implemented, and the downstream tool must fill in the blank information on its own. As such, downstream tools have their own implementation of

collective algorithms and choose from these options as necessary. **To the best of our knowledge, there is no common format that represents both the distributed ML workloads and collective algorithms**. Because of this, collective communications and other operators within a workload (such as compute or memory access) are also optimized separately. **Additionally, the absence of a common representation between upstream producers and downstream consumers limits interoperability.** In order to evaluate custom collective algorithms with a desired simulator, users have to manually program the algorithm following *simulator-specific* rules. This adds a layer of development burden to the search for collective algorithm optimizations.

In order to bridge this gap, **we propose a common representation format for both workload and collective algorithms across different tools**. We draw attention to the Chakra Execution Trace (Chakra ET) [22] format. Chakra ET is a standard graph-based representation of execution traces for distributed ML workloads, and is currently being developed by MLCommons [23]. In this work, we propose to extend Chakra ET to also encode collective algorithms. Fig. 1 summarizes the standardization proposal, where (i) both the distributed ML workload and collective algorithm (produced by the upstream tools) are represented in a common Chakra ET format and (ii) downstream tools can directly ingest a single Chakra ET representation for simulation or execution. We develop a proof-of-concept workflow allowing users to write custom collective algorithms using the MSCCLang Domain Specific Language [18], represent them in Chakra ET format, and compare the custom algorithms using the ASTRA-sim distributed ML system simulator [19]. Similar workflows with different tools can be easily built leveraging this standard representation format.

There are three key benefits to employing Chakra ET as a common collective algorithm representation. First, we envision to **co-optimize collective communication with other operations in the workload via leveraging a streamlined workflow** encapsulating both the workload and collective algorithms. Second, the decoupling between upstream and downstream tools helps **reduce the effort to develop and implement collective algorithms for each downstream tool**. For example, distributed ML simulators like ASTRA-sim [19] could ingest the common representation instead of users having to implement the collective algorithm within its codebase, which requires *simulator-specific* knowledge. Finally, **interoperability across different downstream tools** is another benefit, where users can leverage a single representation to test the same collective algorithm on both simulators like ASTRA-sim and real systems like MSCCL-Runtime [18].

To summarize, our contributions are as follows:
- We motivate and propose a standard workflow that uses a common collective algorithm representation to bridge distributed ML workload information, upstream collective algorithm producers, and downstream tools.
- As proof-of-concept, we provide a case study showing interoperability with MSCCLang and the ASTRA-sim simulator.

## II. BACKGROUND

### A. Chakra Execution Trace

Chakra ET [22] aims to provide a standard graph-based representation to capture the trace of distributed ML workload execution. Chakra ET represents a distributed workload using a directed acyclic graph whose vertex denotes ML operations and edge indicates inter-operation dependency. Each node can be of type COMP, MEM, or COMM_COLL (either collective communication or single COMM_SEND and COMM_RECV). Chakra ET can be fed into and consumed by distinct downstream tools, such as distributed ML simulators or benchmarking tools. These downstream tools leverage Chakra ET by traversing through the graph and issuing the operations whose dependencies are resolved and are ready to be dispatched. Chakra ET graphs can be collected in multiple ways, for instance, through profiling actual PyTorch executions or via synthetic generations.

Note that a collective communication node in Chakra ET merely indicates that a collective communication has taken place. However, it does not encode the actual collective algorithm. In other words, Chakra ET itself is oblivious to exactly how the messages are orchestrated and transferred. Consequently, the downstream tools exploit their internal, tool-specific implementations of collective algorithms.

### B. Upstream Collective Algorithm Producers

Given the significance of collective communication in distributed ML, numerous studies have focused on optimizing collective algorithms. Primarily, they have pursued two main threads: (i) developing domain-specific languages (DSLs) to enable users to define their own collective algorithms and (ii) implementing synthesizers to autonomously generate them.

One notable example is MSCCLang [18], which introduces a Python-based DSL for collective algorithms. This enables users to easily construct NCCL-based collective algorithms. MSCCLang compiles these algorithms into an XML-based representation (MSCCL-IR), which is then executed on real clusters via the NCCL-based MSCCL-Runtime. Meanwhile, synthesizers like TACCL [17] and TACOS [24] generate collective algorithms tailored to the network topology. TACCL employs Integer Linear Programming (ILP) to identify optimal collective algorithms, while TACOS utilizes a Time-expanded Network (TEN) approach.

### C. Downstream Distributed Machine Learning Tools

Downstream tools receive distributed ML workloads or collective algorithm representations to execute meaningful tasks. Collective communication runtimes and distributed ML simulators are notable instances of such downstream tools. For instance, the MSCCL-Runtime orchestrates a collective communication by taking the collective algorithm in MSCCL-IR format (compiled from MSCCLang) and executing it on real GPU clusters via an NCCL-based runtime. Conversely, ASTRA-sim is a notable example of simulation infrastructure. Its Workload Layer can receive and simulate a distributed ML workload represented in Chakra ET format, while its System

TABLE I: A list of Chakra ET node types used to represent collective algorithms and their description.

| Chakra ET Node Type | Description |
|---|---|
| COMM_SEND | (Pt-to-pt) Message send to a destination. |
| COMM_RECV | (Pt-to-pt) Wait for a message from a source. |
| COMP | Execute a compute task on a given hardware (e.g., Reduction operation) |

Layer implements collective algorithms to fill in the gaps of the workload Chakra ET. The Network Layer captures requests from these layers and simulating the actual network transfers over a network simulator of user choice.

## III. COLLECTIVE ALGORITHM REPRESENTATION

### A. Motivation: Needs for Standardization

Currently, upstream collective algorithm producers employ unique representations to describe their results. For example, MSCCLang utilizes a NCCL-based, low-level XML interpretation, while TACOS relies on its own TEN representation.

This lack of standardization often leads downstream tools to rely on their *unique internal implementation*. Consequently, the format and pipeline that downstream tools use to fetch collective information diverge from those used to inject workload information. As a result, users are constrained to optimizing either collective operations or other workload operations, not both.

Moreover, the absence of a standard format means that executing an upstream producer's algorithm with a specific downstream tool requires users to comprehend the internal details of both tools and implement the algorithm themselves. This task is not only highly prohibitive but also implies that it must be repeated for every pair of upstream and downstream tools. Consequently, upstream and downstream processes become disjointed and lack plug-and-play functionality.

### B. Solution: Using Chakra Execution Trace

We standardize the collective algorithm representation by utilizing the Chakra ET format that is already employed for distributed ML workloads. It readily offers mechanisms to capture point-to-point message transfers between arbitrary NPUs as well as compute operations necessary for collectives.

By representing both distributed ML workloads and collective algorithms in Chakra ET, we cleanly resolve the issue of separation among workloads, upstream, and downstream tools. Fig. 1 depicts the proposed standardized workflow. Both distributed ML workloads and collective algorithms are in Chakra ET format and passed on to the downstream. The downstream tool then traverses the workload ET and executes operations as their dependencies are resolved. During the process, since collective communication nodes lack the exact mechanism to execute collectives, the downstream tools must decide the algorithm. Previously, they selected a collective algorithm from a range of native implementations or custom algorithms tailored to their specifications. However, with the capability to receive any collective algorithm in Chakra ET, the downstream tools can easily expand the collective communication node with the provided algorithm. Users can rapidly

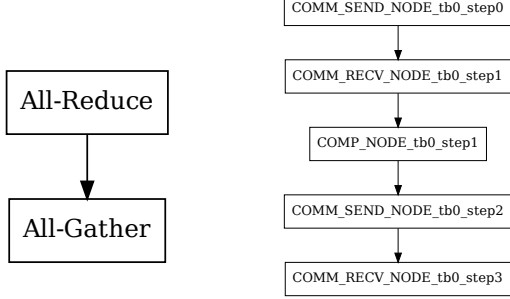

(a) Distributed ML Workload Representation (b) Part of the Collective Algorithm Representation

Fig. 2: Snippets of the Chakra ET used to represent the workload and Ring collective algorithm used in the evaluation.

test new collective algorithms by simply switching out the Chakra ET files without having to create new tool-specific implementations.

By representing collective algorithms with the same format as the workload, we elevate the send and receive messages of a collective algorithm to the same level as other operators in the workload. This opens up opportunities for co-optimizing collective communication and other operators such as compute. For example, it is much simpler to test a scheduling feature that reorders a compute node and a send node (assuming the reorder respects inter-node dependencies) as the compute and send node now use a common Chakra ET format.

Note that real-world systems such as MSCCL-Runtime do not take workload information as input. Even in this case, having the Chakra ET as a common collective algorithm representation helps bridge the gap with upstream producers. For example, a user may want to simulate multiple prospective algorithms using ASTRA-sim, then validate the best-performing candidates by actually using the MSCCL-Runtime. The user in this workflow can reuse the same Chakra ET format across both tools without modification. This is achieved by the fact that the common representation abstracts away the details of the downstream tools.

### C. Collective Algorithm in Chakra ET

Table I lists the types of Chakra ET operator nodes that are used to represent collective algorithms and their definition. Leveraging these nodes, Chakra ET can represent point-to-point network transfer between two NPUs as well as compute operations. Arbitrary collective algorithms can then be described as a combination of network message transfers and reductions.

To meet the standardization requirement, upstream tools need to convert resulting collective algorithms from their default representation to the common Chakra ET format when producing collective algorithms. We highlight that implementing this conversion is a one-time task such that, once developed, can be reused across multiple downstream frameworks.

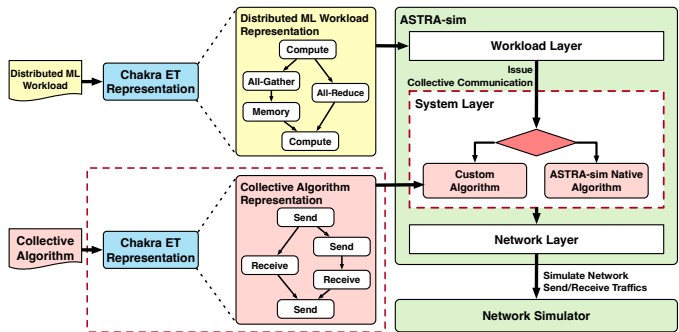

Fig. 3: Components of ASTRA-sim involved in the case study. We extended ASTRA-sim to inject collective algorithms represented in Chakra ET. The extension is marked with dashed squares.

## IV. METHODOLOGY

As a case study and proof-of-concept, we construct an end-to-end workflow using MSCCLang and ASTRA-sim. Here, we describe the extensions made to the two tools for the case study. We expect similar extensions would apply to other tools.

### A. Representing MSCCLang Output in Chakra ET

We developed a converter that bridges the MSCCL-IR format into Chakra ET. The converter creates a vertex (i.e., Chakra ET node) for each operation in the MSCCL-IR format. The converter then creates edges by extracting inter-operator dependency information encoded in the MSCCL-IR format. To showcase the update, we described a 1D Ring algorithm of All-Reduce using MSCCLang and compiled the result in the standard Chakra ET format. Fig. 2(b) shows part of the Chakra ET-based 1D Ring algorithm representation.

### B. Updating ASTRA-sim to Run Algorithms in Chakra ET

Fig. 3 shows the workflow of ASTRA-sim simulator and our extensions to it. ASTRA-sim includes its implementation of collective algorithms found in NCCL, such as Ring or Double Binary Tree, out of the box as part of the program.

At each run, the user will choose which algorithm to use for each collective. Whenever a collective communication node is issued, ASTRA-sim will run the corresponding algorithm code. We extend ASTRA-sim by adding an input parameter for the user to refer to the collective algorithm in Chakra ET format. The simulator will parse the provided Chakra ET and simulate the point-to-point sends and receives following the dependencies, rather than using its own implementation. This process is depicted in dashed boxes in Fig. 3. Since ASTRA-sim readily supports the execution of ML workloads in Chakra ET, reusing the components in the workload layer has made it easy to add the extension for the common collective representations as well.

## V. EVALUATION

We showcase our case study of bridging MSCCLang and ASTRA-sim as described in Sec. IV. We use an All-Reduce collective followed by an All-Gather collective for the workload and use MSCCLang to generate a 1D Ring algorithm for both collectives. Fig. 2 shows a snippet of the Chakra

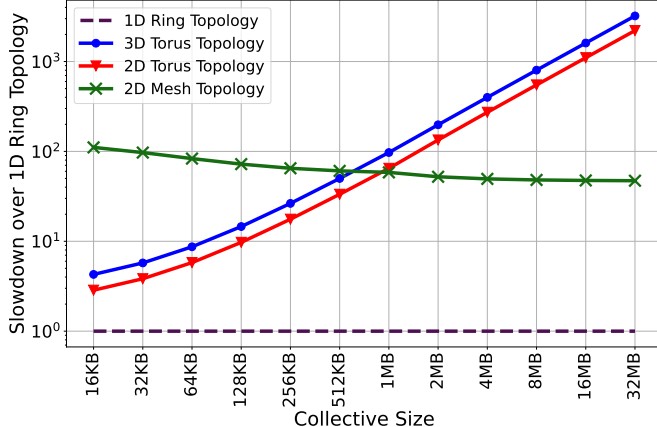

Fig. 4: The collective duration for a 1D Ring algorithm across different topologies of 64 NPUs.

ET represented as a graph. Note how we are able to represent both the workload and the collective algorithm using the same Chakra ET format. The resulting Chakra ET representation is then provided to the ASTRA-sim simulator to be tested across different topologies with varying physical connectivity.

We use the Analytical network simulator to model the message transfer. The topologies consist of 64 NPUs with varying connectivity. We observe the workload duration as we differ the size of the collective. Fig. 4 shows the simulation results, using the slowdown of the different topologies compared to a 1D Ring. Note that such an experiment was made possible thanks to the streamlined workflow via standardization of the collective algorithm representation. It is natural that other topologies show slowdowns as we use a simple Ring algorithm for both All-Reduce and All-Gather. While this evaluation showcases a simplistic workload, it is possible to expand the evaluation to include complex workloads and collective algorithms. We leave this to future work.

## VI. CONCLUSION AND FUTURE WORK

This paper proposes a standardized common representation for collective algorithms. We reuse the Chakra ET format, which already captures distributed ML workload traces, as the collective representation. Representing both workloads and collective algorithms with the same format will allow us to explore the co-optimization of workload operators and collective communication operators. We showcase such a common representation with a case study using collective algorithms produced by MSCCLang on the ASTRA-sim simulator.

Our work opens up several future research directions. One important future work is to leverage the proposed standard representation to explore collective optimizations in the context of an actual workload. We anticipate that a common representation across a workload and a collective algorithm will allow us to study the co-optimization of compute and communication operations. This allows researchers to further study the overlapping of compute and communication operations. Another potential direction would be to expand other tools to produce and consume Chakra ET format, to further expand the ecosystem and scope of our proposed workflow.

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
