# OpenReview forum: "Towards a Standardized Representation for Deep Learning Collective Algorithms"
_iscaconf.org/ISCA/2024/Workshop/MLArchSys — MLArchSys 2024 OralPoster_

### Official Review · Reviewer_ZsYs · 2024-05-29

**Confidence:** 3
**Rating:** 7

**Detailed Feedback And Questions For Authors:**

The quality of the work is high, with a well-defined problem statement and a comprehensive proposed solution. The authors provide a clear explanation of the current challenges in distributed machine learning regarding collective algorithms and their representation. The use of Chakra ET as a standardized representation is well-motivated, and the proof-of-concept implementation demonstrates the practicality of the proposed approach.

The clarity of the paper is generally good, with well-organized sections that guide the reader through the problem, solution, methodology, and evaluation. However, some technical sections could benefit from additional explanations and visual aids to make the concepts more accessible to a broader audience. For instance, the detailed process of converting collective algorithms to Chakra ET format could be accompanied by more illustrative examples.

The originality of the work is significant, as it addresses a critical gap in the field of distributed machine learning. The idea of using a standardized execution trace format to unify the representation of collective algorithms and workloads is novel and has the potential to impact both academia and industry positively.

The significance of the work is high. By providing a standardized representation, the authors enable more seamless integration and interoperability between different tools and platforms used in distributed machine learning. This can lead to more efficient development, testing, and optimization of collective algorithms, ultimately advancing the state-of-the-art in distributed ML systems.

**Top Reasons To Accept The Paper:**

* The paper addresses a critical gap in distributed machine learning by proposing a standardized representation for collective algorithms, which can significantly enhance interoperability and co-optimization of compute and communication operations.
* The use of Chakra Execution Trace (Chakra ET) as a common format for representing both distributed ML workloads and collective algorithms is a novel and well-motivated solution that can streamline the development and optimization process.
* The proof-of-concept implementation with MSCCLang and ASTRA-sim demonstrates the feasibility and practical benefits of the proposed approach, providing a solid foundation for future research and development in this area.

**Top Reasons To Reject The Paper:**

* The evaluation section is limited to a specific proof-of-concept and does not cover a wide range of real-world scenarios or more complex workloads, which may limit the generalizability of the findings.
* The paper may be challenging for readers who are not already well-versed in distributed machine learning and the specifics of collective algorithms, potentially limiting its accessibility and impact.
* The paper does not fully address the effort and challenges involved in extending other existing upstream and downstream tools to support the proposed Chakra ET standard, which may pose practical barriers to adoption.

---

### Official Review · Reviewer_PJyz · 2024-05-29
**Towards a Standardized Representation for Deep Learning Collective Algorithms**

**Confidence:** 5
**Rating:** 6

**Detailed Feedback And Questions For Authors:**

## Summary
* This paper tries to address the problem that the lack of a unified representation for compute operations
and collectives in distributed ML serving use cases leads to suboptimal implementations.
* The paper proposes to extend and use the Chakra execution trace to bridge this gap.
* The paper provides a proof of concept case study where the same representation is used by collective optimization
algorithm and ML partitioning algorithm to co-optimize execution

## Comments
Thank you for submitting your work to MlArchSys. I liked the motivation for using the same intermediate
representation, however I think the paper needs some more evaluation to help readers understand the magnitude
of the contributions. Please try to address the following in any subsequent drafts:
1. Please explain why chakra ET is vital. For example, each tools can use its own representation but the optimization can
be done iteratively and collaboratively to attain an optimized execution configuration. Please highlight
what a common representation enables as compared to this collaborative approach.
2. Please showcase some salient points of the chakra ET representation that makes it useful for both the upstream and downstream tools.

**Top Reasons To Accept The Paper:**

1. The presented concept of using a common representation format for collectives for compute can help
co optimize distributed ML serving.

**Top Reasons To Reject The Paper:**

NA

---

### Official Review · Reviewer_DieT · 2024-05-29
**Review: Towards a Standardized Representation for Deep Learning Collective Algorithms**

**Confidence:** 4
**Rating:** 7

**Detailed Feedback And Questions For Authors:**

As you paper mentions, there are 3 primary reasons for a standardized representation:

(1) Reduce the effort to develop collective algorithms for downstream tools.

(2) Interoperability across different downstream tools.

(3) Co-optimize collective communication with other operations in the workload.

(1) and (2) are valuable and there is no debating that they are necessary to enable more streamlined and effective systems research on this topic.

However, (3) was what I found to be the most intriguing and compelling as a researcher. This potential benefit was mentioned multiple times in the paper and I kept waiting as a reader for the actual demonstration of this co-optimization piece. Eventually I finished the evaluation and found that it will be part of the future work. I’m excited to see how these results turn out as it was something I was wanting to see in the case study. I think adding this piece will strengthen the contributions needed to take the next step in this work towards a research conference submission. Overall, I thought the paper was well-written, making it easy to read and follow.

**Top Reasons To Accept The Paper:**

This paper should be accepted as it highlights and identifies a gap that the community needs to be made aware of regarding DL collective algorithms (which is perhaps a bit overlooked at the moment), while providing a viable solution for standardization. Given the explosion in large-scale distributed ML training, I feel this is a very timely and strong call to action. Standardization and interoperability is significant and a topic that the community needs to prioritize.

**Top Reasons To Reject The Paper:**

None.

---

### Decision · Program_Chairs · 2024-05-30

**Decision:**

Accept (Oral/Poster)

**Comment:**

Congratulations! We are pleased to inform you that your paper has been accepted for presentation at MLArchSys 2024. We look forward to your participation at the workshop. Further details regarding the schedule and format will be provided soon. See you at the workshop!